# Asian summer monsoon variability across Termination II and implications for ice age terminations

Yijia Liang[1,2], Kan Zhao [1] ✉, Yongjin Wang [1] ✉, Shitao Chen[1], Tyler E. Huth[3,4], Bin Zhao[1], Quan Wang[5], Zhenqiu Zhang[1], Qingfeng Shao [1], Hai Cheng [6] & R. Lawrence Edwards[7]

The detailed anatomy of Termination I (TI) is well depicted, but whether changes across Termination II (TII) resemble TI remains controversial. Here we present high-resolution Asian monsoon records covering TII using Shima Cave stalagmites from China. Correlating marine and ice-core records to our U/Th-dated records via millennial-scale variabilities, we find an initial $CO_2$ rise from $139 \pm 1$ ka BP concordant with boreal summer insolation increase, which was followed by a major rise phase of $CO_2$ between $135.7 \pm 1$ and $129 \pm 1$ ka BP. The major rise phases of $CO_2$ were comparable during TI and TII, but the initial $CO_2$ rise before TII was distinct from $CO_2$ behavior before TI, likely forced by the Earth's internal variabilities, in particular an ice-sheet collapse event and a 50% reduction in southern hemisphere dust flux. Here, we show that ~4000–5000-year-long gradual changes in $CO_2$, along with insolation rise, preconditioned glacial terminations, supporting the "tipping point" theory.

Quaternary (≈2.6 Ma to present) climate was characterized by a series of alternations between warm interglacial and cold glacial. Records of glacial-interglacial change typically show a "sawtooth" pattern, where gradual ice sheet accumulation and sea level reduction during glacial periods were followed by ice age terminations with rapid ice sheet melting and sea level rise[1–3]. In an early study by Broecker and Denton[4], glacial terminations were ubiquitously associated with millennial-scale oscillations in ocean circulation and ultimately inferred as the cause of $CO_2$ release into the atmosphere. Several subsequent studies support the views of Broecker and Denton[4], in particular identifying the critical role of the Atlantic Meridional Overturning Circulation (AMOC)[5–8]. During ice age terminations, one or two weak monsoon intervals (WMIs) occurred, coinciding with Heinrich (H) events characterized by massive iceberg discharges, where $CO_2$ rise could be caused via a set of

mechanisms ultimately linked to increasing boreal summer insolation[9]. Hence, Asian summer monsoon records provide insights into the feedback and interactions between millennial- and orbital-scale processes.

Current understanding of ocean-atmosphere-cryosphere interactions during ice age terminations mainly relies on the assessment of Termination I (TI, ~18–11.7 thousand years before the present, hereafter ka BP) due to the abundance of archives with reliable, independent age constraints[5]. Denton et al.[5] suggested that the collapse of northern hemisphere ice sheets, due to rising insolation, disrupted the global oceanic and atmospheric circulations, leading to $CO_2$ release from the Southern Ocean that further augmented global warming during the last termination. However, TI may not be representative of previous glacial terminations. An increasing body of evidence from

[1]State Key Laboratory of Climate System Prediction and Risk Management, Jiangsu Center for Collaborative Innovation in Geographical Information Resource Development and Application, School of Geography, Nanjing Normal University, Nanjing, China. [2]School of Geographical Science, Nantong University, Nantong, China. [3]Department of Earth, Environmental, and Planetary Sciences, Washington University, St. Louis, MO, USA. [4]Department of Earth and Environmental Sciences, University of Michigan, Ann Arbor, MI, USA. [5]Research Centre for Environmental Change and Sustainable Development, School of International Business and Tourism Management, Ningbo Polytechnic, Ningbo, China. [6]Institute of Global Environmental Change, Xi'an Jiaotong University, Xi'an, China. [7]Department of Earth and Environmental Sciences, University of Minnesota, Minneapolis, MN, USA. ✉e-mail: 09371@njnu.edu.cn; yjwang@njnu.edu.cn

high-resolution ocean and terrestrial records suggests climatic similarities and dissimilarities between TI and TII, where TII is the penultimate ice age termination (TII, ~136–129 ka BP), containing the transition from the Penultimate Glacial Maximum (PGM, ~140 ka BP) to the Last Interglacial (~129–116 ka BP)[6,10–18]. While the massive continental ice sheets disintegrated as the Heinrich (H) event 11 during TII[11,19], no clear evidence of a Younger Dryas-like event in the North Atlantic sea surface temperature (SST)[14] nor a Bølling-Allerød-like event in the AMOC intensity occurred during TII[6]. Importantly, Cheng et al.[9] observed that monsoon structure during TII was different from TI, lacking the Bølling-Allerød period. Another discrepancy was the relatively higher atmospheric $CO_2$ concentration by >15 ppm at the end of TII than that of TI[20]. Accordingly, Barker and Knorr[8] suggested that Denton et al.'s model[5] could not explain the anatomy of all ice age terminations. To better understand why the two most recent ice age terminations exhibited different characteristics, especially the rise in atmospheric $CO_2$ which plays a critical role in regulating the Earth's temperature, we must investigate the cascade of climatic elements and dynamics before and during the ice age terminations.

In this work, we present a chronological benchmark for the linkage between major ice-rafted debris (IRD) peaks, variations of atmospheric $CO_2$ and weak monsoon shifts across TII based on four high-resolution and U/Th-dated speleothem records from Shima Cave in China. Particularly, we focus on defining the preconditions during the glacial maximum, which might be critical in initiating $CO_2$ release across TII as a way to understand the roles of external solar forcing and internal variabilities in the ocean and atmosphere system during ice age terminations.

## Results and discussion
### Stalagmite samples
Four broken stalagmite samples SM9, SM12, SM16 and SM17 were collected from Shima Cave, Hunan Province, central China (29°35′N, 109°31′E, 650 meters above sea level) (Supplementary Fig. 1). Shima Cave was developed in the Permian limestone and is strongly influenced by the Asian monsoon system. Modern temperature, precipitation, and monthly simulated precipitation $\delta^{18}O$ data all show seasonal variations (Supplementary Fig. 1). The mean annual temperature is ~17 °C and the mean annual precipitation is ~1430 mm, with nearly 57% of the precipitation occurring from May to August when abundant low-$\delta^{18}O$-valued moistures are sourced from remote tropical oceans[21].

A total of 26 $^{230}Th$ dates for four samples are presented in Supplementary Table 1. Most subsamples have low $^{232}Th$ content and high $^{230}Th/^{232}Th$ activity ratios, leading to small, <100 years, initial detrital $^{230}Th$ corrections and dating uncertainties of 300–500 years. The age models for Shima samples were obtained using the MOD_AGE model[22] (Supplementary Fig. 2) because some dates were reversed within dating errors. Only one age outlier exists in SM9 sample, younger than the modeled age by approximately 600 years. According to age models, SM9, SM12, SM16 and SM17 deposited from 126.7 ± 0.8 to 122.6 ± 0.9 ka BP, 131.7 ± 0.7 to 130.8 ± 0.6 ka BP, 142 ± 1 to 133.6 ± 0.6 ka BP and 129 ± 0.9 to 126.2 ± 0.8 ka BP (BP represents 1950 AD), respectively (Fig. 1a).

A total of 1949 $\delta^{18}O$ subsamples were analysed and used for proxy reconstruction. In addition, five subsamples of stalagmite SM7 from our previous study of TI at Shima Cave[21], were analysed for triple oxygen isotope composition ($\Delta^{47}O$) because SM7's total $\delta^{18}O$ range of 5‰ and the record's completeness over the glacial termination was conducive for defining trends in $\Delta^{47}O$ and $\delta^{18}O$ space (as per meg/‰) that are linked to the driving hydrologic processes[23–25].

### Interpretive basis and speleothem oxygen isotope records
The Shima speleothem $\delta^{18}O$ records are shown in Fig. 1a, and, based on prior work, we interpret them in the context of the intensity of the

Asian summer monsoon (ASM), reflecting the integrated rainfall from the Pacific and Indian Ocean moisture sources to the cave[21,26,27]. Temporal resolution for the record is better than 20 years, with chronologies constrained by most $^{230}Th$ dates with uncertainties <400 years for the section younger than 130 ka BP, and 400–500 years for the older part. To alleviate the temporal gap of the abrupt Asian monsoon TII in the Shima record, we combined our records with that of Sanbao Cave (270 km distant, Supplementary Fig. 1), which is well-dated and has dating errors of less than 100 years[9] (Fig. 2c), by adding a systematic bias of 0.2‰ to SB25 $\delta^{18}O$ data. We can accordingly precisely determine the timing of millennial-scale events evident in ice cores and marine sediments by correlating them to the radiometrically dated cave $\delta^{18}O$ record (Fig. 2c).

Calcite precipitation in isotopic equilibrium is a prerequisite for using its $\delta^{18}O$ as a climate proxy. Three lines of evidence suggest that Shima $\delta^{18}O$ records reflect formation in isotopic equilibrium. First, the Hendy test[28] was performed on four individual growth layers of each sample. Most $\delta^{18}O$ and $\delta^{13}C$ variations along the same layer range from 0.1‰ to 0.3‰ (Fig. 1b), and the absence of positive relationships between $\delta^{18}O$ and $\delta^{13}C$ are consistent with the equilibrium formation of calcite (Supplementary Fig. 3). Second, replication of different stalagmite $\delta^{18}O$ records is reliable to test for equilibrium fractionation in carbonates[29]. Our Shima records display good consistency with other cave records from southern China on the orbital to millennial timescales, including abrupt positive shifts of ~2‰ at approximately 136 ka BP and negative shifts of >4‰ from the TII to the Last Interglacial (Supplementary Fig. 4). Third, the $\Delta^{47}O$ vs. $\delta^{48}O$ trend of Shima stalagmite SM7 has a slope of ≈ 1 per meg/‰ (Fig. 1c), consistent with the predominance of rainout processes in determining $\delta^{18}O$. Similar trends could mathematically come from combinations of multiple processes[24]. However, even combining trends from, for example, rainout (Rayleigh, 0 per meg/‰) and within-cave kinetic (7 per meg/‰) processes would still require an overwhelming 86% of the signal to be coming from climate-related drivers (i.e., 0.86 × 0 per meg/‰ + 0.14 × 7 per meg/‰ = 1 per meg/‰). Changes in $\Delta^{47}O$ can also come from varying the relative humidity during evaporation at the moisture source (−9.5 per meg/‰) and require only a 37.5% contribution from the climate signal (i.e., 0.375 × (−9 per meg/‰) + 0.625 × 7 per meg/‰ = 1 per meg/‰). These calculations, in combination with the fact that modern precipitation in the Asian monsoon region shows weak, slightly negative to slightly positive trends (Supplementary Fig. 5), also support stalagmite formation in (near-)isotopic equilibrium. Ultimately, while these lines of evidence are useful in establishing that the Shima samples formed in (near-)isotope equilibrium, our fundamental interpretations below are about the timing of change, not the exact interpretation of speleothem $\delta^{18}O$, and thus independent of the driving processes.

### Timing of the initial rise of $CO_2$ during the Penultimate Glacial Maximum
Based on a mechanical link of abrupt shifts in ASM (under whose influence the wetland is one of the major sources for atmospheric $CH_4$) and $CH_4$ in Antarctic ice cores during ice age termination[9,30], we compare the Shima-Sanbao record with $CH_4$ and $CO_2$ records from the EDC ice core on the AICC2012 chronology[31,32]. Apart from the nearly-consistent abrupt intensification/rise in ASM and $CH_4$[33] at around 129 ka BP (blue dashed line in Fig. 2b, c), a small peak of $CH_4$ at ~134 ka BP is possibly tied to a short-lived strengthening of the ASM, which has been confirmed in a previous study[34] and further constrains the chronology of $CO_2$ and $CH_4$ records for older sections. All three atmospheric $CO_2$ records[17,20,31,35] began to rise at ~140 ka BP until 129 ka BP, lasting for ~11,000 years with a total rise of ~100 ppm (Fig. 2a). The onset timing of $CO_2$ rise cannot be determined more precisely due to the low-resolution data of $CO_2$ at ~140 ka BP. However, considering that: (i) the linkage of North Atlantic IRD events or H stadials with rises of $CO_2$

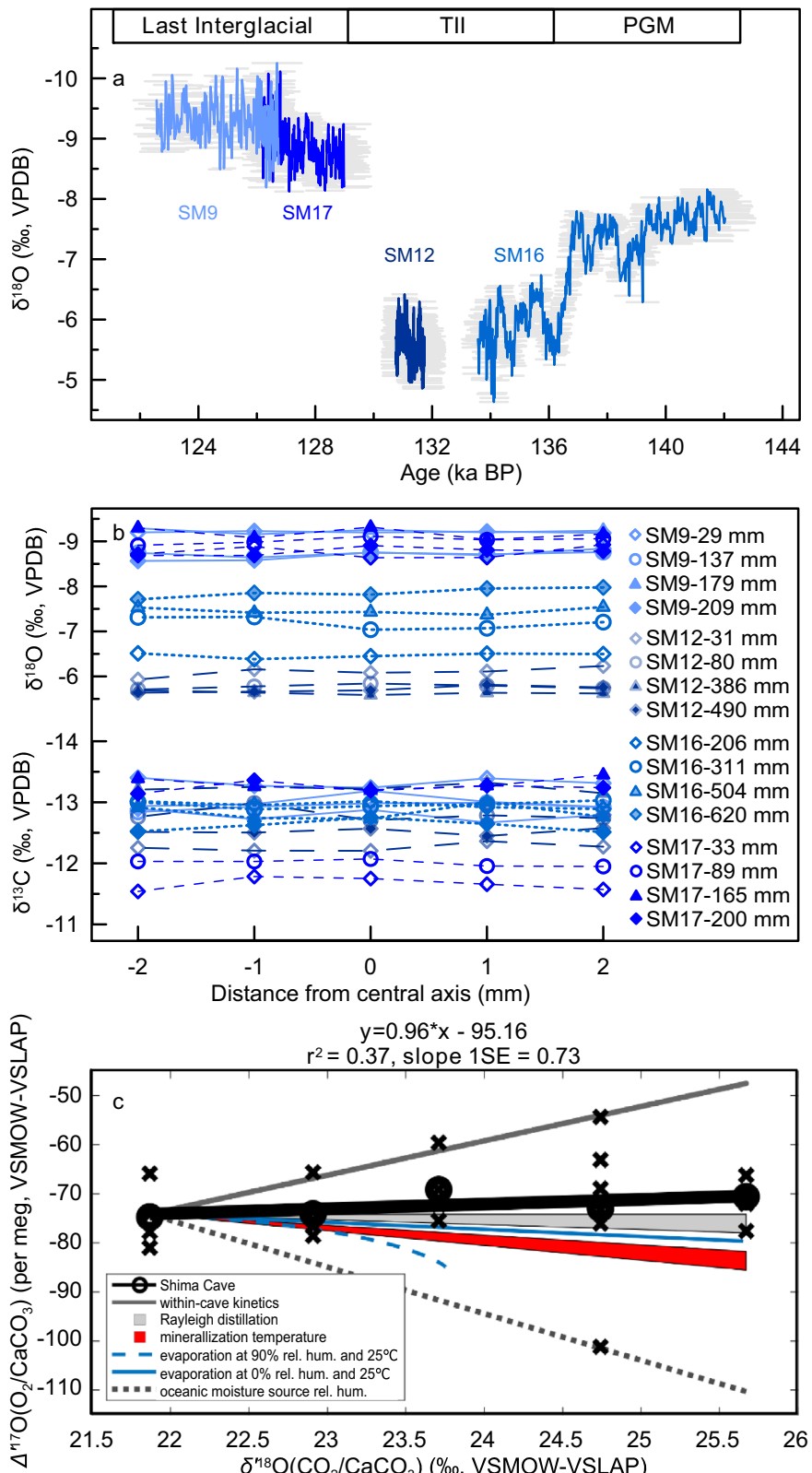

**Fig. 1 | Speleothem stable oxygen data from Shima Cave and cave kinetic tests.**
**a** Age model and reconstructed $\delta^{18}O$ records with sample ID. Grey shadows are uncertainties of modeled chronology. **b** Hendy test: $\delta^{13}C$ and $\delta^{18}O$ results for individual layers, with different symbols and colors indicating different depths in four samples. **c** Shima Cave $\Delta^{47}O$ vs. $\delta^{48}O$ data (averages: black dots, replicates: black crosses) and trend (bold black line) compared to characteristic trends for hydrologic processes: within-cave kinetics (solid dark grey), Rayleigh distillation at 25 °C (medium grey polygon), pan evaporation endmembers (solid and dashed blue), mineralization temperature (red polygon), and oceanic moisture source relative humidity (black dotted) (rel. hum. relative humidity). TII Termination II, PGM Penultimate Glacial Maximum. Source data are provided as a Source Data file.

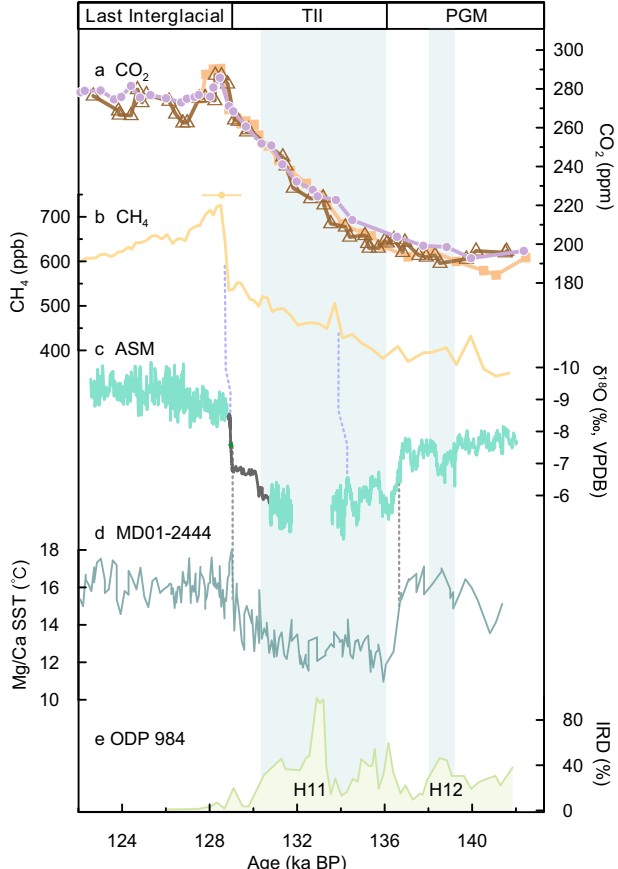

**Fig. 2 | Comparison of paleoclimate records across Termination II. a** $CO_2$ records from the EDC ice core (orange rectangles[35], brown triangles[20], purple dots[17,31]), and **b** $CH_4$ record from the EDC ice core[33], all on the AICC2012 gas-age chronology[32]. The yellow dot and error bar indicate abrupt $CH_4$ and $CO_2$ increase with an error from the AICC2012 chronology[32]. **c** A high-resolution and well-dated Asian summer monsoon (ASM) record combining Shima records (green, this study) with the Sanbao record[9] (black). The green dot and error bar indicate abrupt ASM intensification with a dating error of 100 years from Sanbao Cave record[9]. **d** North Atlantic sea surface temperature (SST) record from core MD01-2444[36]. **e** Ice-rafted debris (IRD) percentage from core ODP984[37] on the tuned chronology[36]. Green bars denote the Heinrich (H) events 12 and 11. Blue dashed lines link $CH_4$ record with the ASM, and black dashed lines link SST record with the ASM. TII Termination II, PGM Penultimate Glacial Maximum. Source data are provided as a Source Data file.

through a mechanism of the AMOC weakening that promotes the release of $CO_2$ from the Southern Ocean to the atmosphere[5] and (ii) good correlations of WMIs and IRD events[9,21,26], here we can provide constraints on the timing of initial $CO_2$ rises by bridging the ice core, marine and Shima-Sanbao cave records.

There is a broad similarity between our cave record and the North Atlantic SST record at the site of MD01-2444[36]. The intensive monsoon weakening and strengthening, indicated by >2‰ changes in $\delta^{18}O$ values, were associated with a cooling of ~4 °C at around 136 ka BP and large-amplitude warming at the end of TII in the North Atlantic (black dashed lines in Fig. 2c, d). The WMI at around 139 ka BP, as evidenced by a positive shift of 1.3‰ in Shima $\delta^{18}O$ record and also observed in Hulu, Sanbao and Dongge records (Supplementary Fig. 4), was coherent with a ~1 °C cooling event in the North Atlantic (Fig. 2d). Since the chronologies for ODP 984 and MD01-2444 cores were well aligned by tie points[36], major IRD peaks of H11 and H12 events in core ODP 984[36,37] (Fig. 2e) could also be well linked to the sequence of WMIs in the cave record. Obvious peaks for H12 event in other IRD records on independent age models varied between 140 and 138.5 ka BP, and H11

event showed a more complex structure with its onset at approximately 136 ± 1.5 ka BP (Supplementary Fig. 6). These two major IRD peaks were consistent with AMOC weakening, and tightly correlated with WMIs (Supplementary Fig. 6). Based on the evidence presented, we propose that H12, aligning to a WMI event dated at 139 ± 0.6 ka BP, possibly contributed to the initial rise of $CO_2$ through the weakening or shutdown of the AMOC[5,9] (Supplementary Fig. 6e, f). The timing of WMI in response to H12 during the PGM in the Shima record is consistent with other independently dated Chinese cave records within their dating errors (Supplementary Fig. 4). By evaluating $^{230}Th$ age control for cave records (Supplementary Fig. 4) and the original age model for ice cores[31,32], we suggest that the initial rise of $CO_2$ for TII occurred at 139 ± 1 ka BP.

## Establishing the analogy of the last two ice age terminations

The existing analogy of ice age terminations has been proposed[6,8,9], and these hypotheses involve millennial-scale internal variabilities in the Earth's climate system that set up deglaciations. Here we invoke similar processes in $CO_2$ rise related to monsoon intensity changes to establish an analogy between TI and TII, since $CO_2$ functions as an essential amplifier of global temperature[38] and an important tipping forcing for ice age terminations[39].

We applied the Change Point detector in software Acycle 2.8 to the EDC $CO_2$ record[31] and obtained an abrupt rise of $CO_2$ at 17.4 ± 0.2 ka BP during the early last deglacial period (Supplementary Fig. 7). This result is supported by the determination of the onset of $CO_2$ rise in ref. 38. Following the same procedure, we determined the start of the major rise phase of $CO_2$ during TII to be 135.7 ± 2 ka BP (Supplementary Fig. 7). Considering that the tie points of $CH_4$ jump and fast monsoon recovery in the cave record can constrain the timing of $CO_2$ record as discussed above, we narrowed the uncertainty from ±2 to ±1 ka.

After obtaining the onsets of rapid $CO_2$ rises for the last two ice age terminations, we established their analogy (Fig. 3). Changes in monsoon intensity inferred from cave records and Antarctic temperature inferred from ice $\delta D$ records are comparable. Although the internal sequences of monsoonal events are distinguished between TI and TII[9], analogous changes in cave records exist, including: (i) similarly abrupt weakening and fluctuations in the ASM occurring at the transitions from glacial maximums to TI/TII (green bar in Fig. 3), and (ii) a rapid monsoon intensification after the long-term WMIs which marks an end of ice age termination (yellow bar in Fig. 3). The onset and end of TI/TII bracketed the major rise phase of $CO_2$, which lasted ~6000–7000 years. The TI and TII shared a similarly changing amplitude of ~80 ppm during the major rise phases of $CO_2$, regardless of different substages within them (Fig. 3a). Besides, Antarctic $\delta D$ records[40] fluctuated around −440‰ during both glacial maximums and then took similar amounts of time through deglacial processes to reach their interglacial plateaus, despite the interruption of Antarctic Cold Reversal in TI (Fig. 3b). Landais et al.[41] reported that atmospheric $CO_2$ concentrations and Antarctic temperature started increasing in phase around 136 ka BP, supporting the result of 135.7 ± 1 ka BP here. The beginning of the rapid $CO_2$ rise at around 135.7 ± 1 ka BP was also widely consistent with abrupt changes in a number of oceanic and terrestrial records like the H11 event, cooling in the North Atlantic SST and European surface temperature, as well as the AMOC weakening (Supplementary Fig. 6). Therefore, abrupt shifts in different archives at 135.7 ± 1 ka BP could be critical changes in the global climate system, which shared similar features with the onset of TI[10,14,42,43].

## Different routes to ice age terminations

While TI and TII have similarities in their initiations and overall $CO_2$ increment, distinct patterns of $CO_2$ change are observed before ice age terminations. During the PGM, $CO_2$ already increased gradually from 139 ± 1 to 135.7 ± 1 ka BP, leading to a total rise of 15–20 ppm

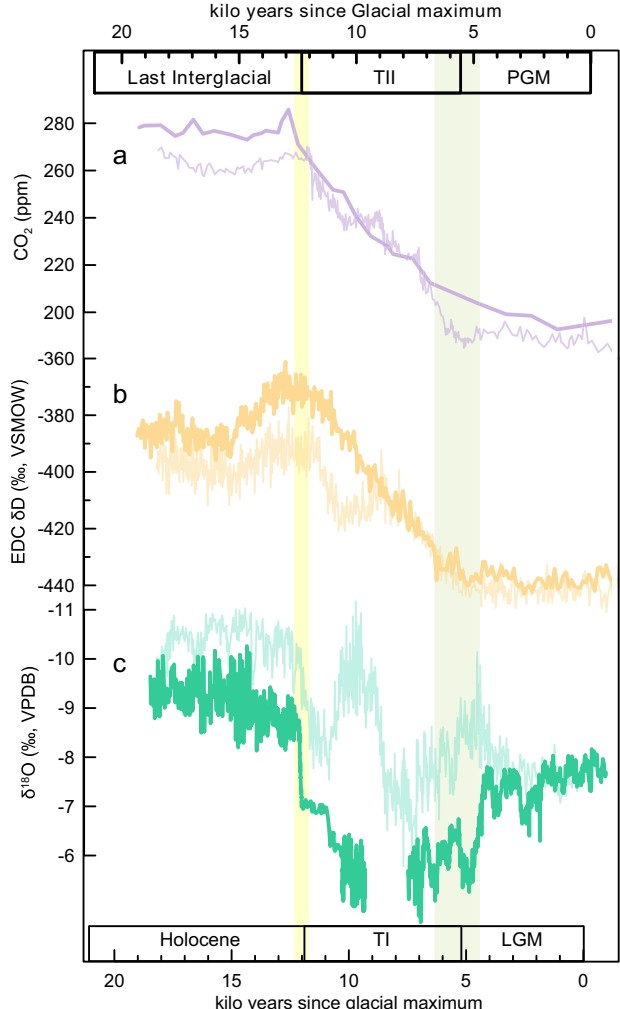

**Fig. 3 | Comparison of Asian summer monsoon, atmospheric CO₂ and Antarctic temperature changes across Termination I (TI) (slim lines) and Termination II (TII) (thick lines).** **a** Composite CO₂ record for TI and TII[31] on the AICC2012 chronology[32]. **b** Antarctic δD record indicating temperature from EDC[40] on the AICC2012 chronology[32]. **c** Composite stalagmite δ¹⁸O records across TI[26] and TII (the same as Fig. 2c). Green and yellow bars indicate the onset and termination of TI/TII, respectively, and bracket the major rise phase of CO₂. LGM Last Glacial Maximum, PGM Penultimate Glacial Maximum. The X-axis is kilo years since glacial maximum. Source data are provided as a Source Data file.

before the abrupt increase (Fig. 4c). However, atmospheric CO₂ concentration fluctuated steadily around 190 ppm for about 5000 years before the TI onset (Fig. 4C). Both isolation and ice volume provide critical backgrounds for CO₂ variations during glacial-interglacial cycles[44], but they cannot explain different CO₂ behaviors during glacial maximums. Because northern hemisphere summer insolation[45] and global ice volume[2] during the LGM (Fig. 4A, B) were broadly similar to those during the PGM (Fig. 4a, b), with insolation on the rising limb from minimums and lasting 4000–5000 years, while LR04 δ¹⁸O values varying around 5‰. Instead, the internal elements of the Earth's climate system might account for differences in CO₂ variations.

We notice three significantly different conditions during the PGM and LGM: (i) a more unstable monsoon status during the PGM than LGM, (ii) the meltwater pulse (MWP) event 2A[11] associated with the H12 occurring at ~139 ka BP[46], and (iii) a significant drop in dust flux recorded in the Antarctic ice cores[47]. During the LGM, the high-resolution speleothem record[26] displays a monsoon intensification

trend with centennial-scale variations (Fig. 4G). However, during the PGM, a significant 139 ± 0.6-ka WMI is detected in addition to centennial oscillations (Fig. 4g). This WMI was likely tele-connected with the H12 event and the MWP-2A (Fig. 4d, e), as well as the North Atlantic SST cooling (Supplementary Fig. 6). The H12 event occurred around the boreal summer insolation minimum (Fig. 4a), and created small, partially deglaciated ice sheets prior to TII[46]. Alike the H11 event which caused an intensive reduction of the northern ice sheets and the MWP-2B[11], the H12 event could have caused the MWP-2A. Although not as intensive as MWP-2B which contributed ~70% of deglacial sea level rise[11], MWP-2A was the early phase of ice-sheet retreat before TII and contributed about 30 meters of sea level rise[48,49]. The concurrence of the H12 event, MWP-2A, the WMI and the North Atlantic cooling episode (Figs. 2, 4) supports a clear millennial-scale climatic oscillation at around 139 ka BP.

More importantly, the 50% off in dust flux observed at ~139 ka BP (Fig. 4f) could have also contributed to CO₂ rise and ultimately arose from the less exposure of the South American continent due to sea level rise and a relatively humid South America. On millennial time-scales, the weakening of the ASM is generally coincident with the strengthening of the South American summer monsoon (SASM)[50], and South America, especially Patagonia, is the primary dust source for Antarctica[51]. At around 139 ka BP, the decrease in dust input to Antarctica and the Southern Ocean might be due to relatively humid conditions[50] and the well-developed vegetation[52] in South America, both associated with a relatively strong SASM status which was anti-phase with the ASM intensity. Increased input of Fe-bearing atmospheric dust to the Fe-deficient Southern Ocean may have stimulated the primary productivity of phytoplankton and enhanced oceanic sequestration of CO₂ during glacial periods, in contrast to changes observed during interglacial periods[53]. It was estimated that ~40 ppm of the change in CO₂ concentration during the glacial-interglacial transitions could be caused by changing dust export to the Southern Ocean[54]. Therefore, a combination of dust flux decreases and the H12 event could possibly cause the initial CO₂ rise, leading to different CO₂ change patterns during the LGM and PGM.

The 15–20 ppm CO₂ increase before 135.7 ± 1 ka BP might explain why atmospheric CO₂ concentration at the end of TII was >15 ppm higher than that at the end of TI. This initial, gradual rise of CO₂ can be regarded, according to the "tipping point" theory[39], as an early warning signal that leads to the inevitable CO₂ rise at the onset of TII. From the viewpoint of physical mechanisms, the abrupt rise of CO₂ since 135.7 ± 1 ka BP was likely due to the increased carbon storage in the stratified Southern Ocean during the preceding PGM[55]. The early warning signal in CO₂ implies that a slow, initial process is important for the shift from glacial to interglacial conditions[39,56], which might operate through a way similar to that proposed by Zhang et al.[57]. By defining the onset of the last two ice age terminations, we find that the major deglacial processes did not occur until ~4000–5000 years after the insolation minimums (Fig. 4), and therefore we suggest that a gradual increase in CO₂ following boreal summer insolation rise could precondition the ice age terminations, thus supporting the "tipping point" theory.

## Methods
### ²³⁰Th dating method
Stalagmites were halved and polished along the growth axis. All samples have a "candle" shape and are composed of pure and transparent calcite. We did not identify deposition hiatuses in these samples, thus indicating continuous deposition (Supplementary Fig. 2). Dating work was performed at the University of Minnesota and Nanjing Normal University, following standard procedures for U and Th separations[58]. Twenty-six powdered samples for ²³⁰Th dating were obtained by drilling along the stalagmite growth axis with a 0.9 mm carbide dental

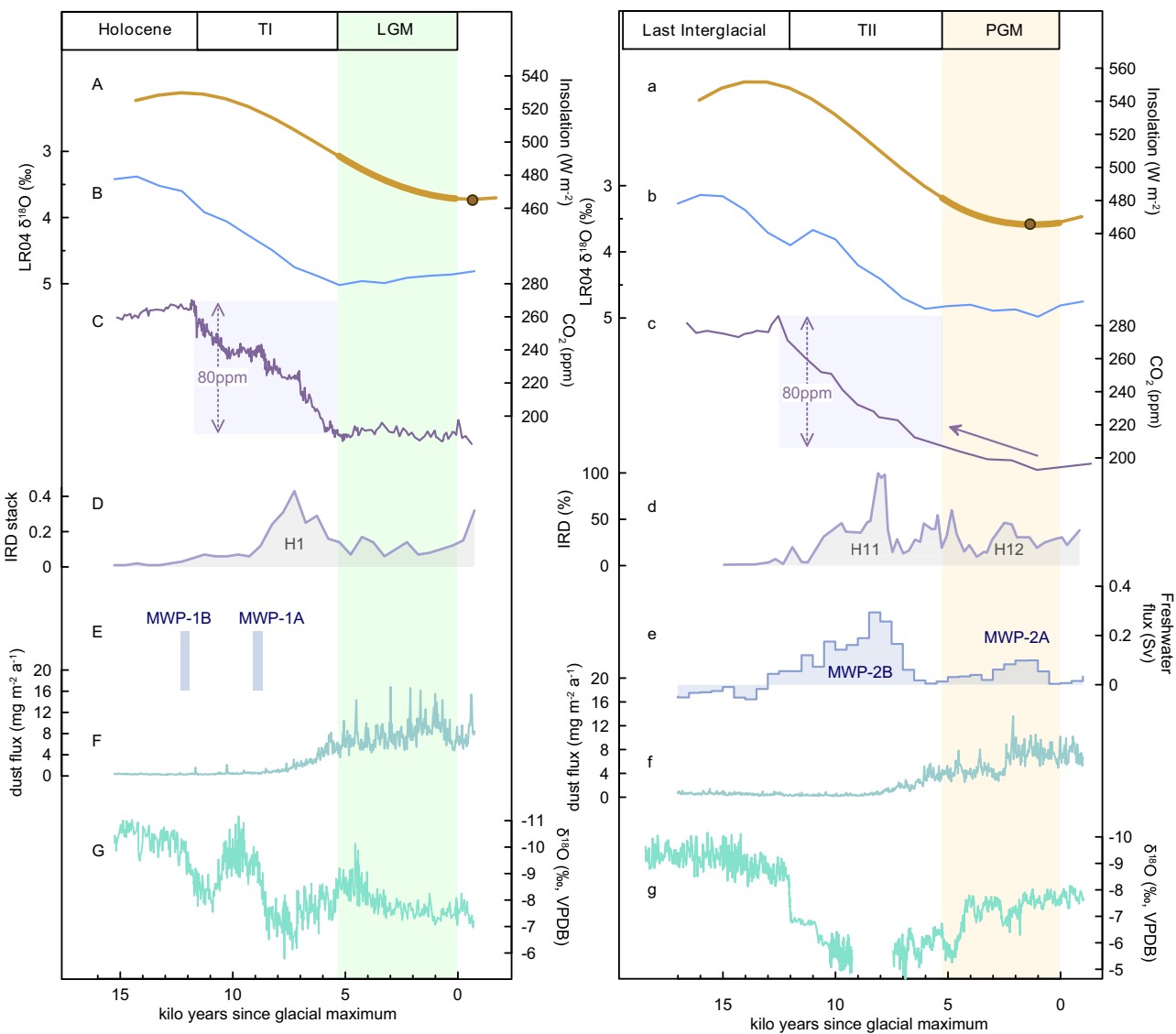

**Fig. 4 | Different preconditions for Termination I (TI) and Termination II (TII).** **A**, **a** 21st June insolation at 65°N[45], with brown dots indicating insolation minimums. **B**, **b** A stacked benthic δ¹⁸O record of LR04[2]. **C**, **c** Composite CO₂ records for TI and TII[31] on the AICC2012 chronology[32], and arrow indicates the increasing trend. Two purple rectangles indicate the major rise phase of CO₂ by ~80 ppm. **D**, **d** Ice-rafted debris (IRD) records from the North Atlantic[36,37,46] and the denoted Heinrich (H) events. **E** Timing of meltwater pulse events (MWP) 1 A and 1B[68] and (**e**) MWP events 2 A and 2B[11]. **F**, **f** Antarctic dust flux record[47] from EDC ice core. **G**, **g** Stalagmite δ¹⁸O records across TI[26] and TII (the same as Fig. 2c). Green and yellow bars indicate the Last Glacial Maximum (LGM) and the Penultimate Glacial Maximum (PGM). The X-axis is kilo years since glacial maximum. Source data are provided as a Source Data file.

drill. They were then weighed and dissolved in HNO₃ in Teflon beakers containing a known quantity of a ²²⁹Th-²³³U-²³⁶U triple spike. U and Th were preconcentrated by coprecipitation with iron hydroxide and then separated by passing the solution through a resin column. The U and Th fractions were then dried and diluted in a mixture of 1% HNO₃ and 0.05% HF for analysis on a Neptune multi-collector inductively coupled plasma mass spectrometer (MC-ICP-MS). The U and Th isotopic data were acquired by the secondary electron multiplier protocol and then processed in the Excel spreadsheets or an interactive program written with the MATLAB programming language[59,60]. Corrections for instrument memory, instrumental mass fractionation, tailing effects, spike contributions, abundance sensitivity, dark noise, and blanks were performed during data processing offline. Decay constants of ²³⁴U[61], ²³⁰Th[61] and ²³⁸U[62] were used. Corrected ²³⁰Th ages assume the initial ²³⁰Th/²³²Th atomic ratio of (4.4 ± 2.2) × 10⁻⁶, and those are the values for a material at secular equilibrium, with the bulk earth ²³²Th/²³⁸U value of

3.8. Most speleothem ages are in stratigraphic order with 2σ analytical errors of roughly 0.1–0.6% (Supplementary Table 1).

### Stable isotope analyses

A total of 1885 subsamples from Shima stalagmites were drilled along the growth axis with 0.5 mm carbide dental burs for stable isotope analysis. And 64 subsamples for the Hendy test were drilled on four individual growth layers in each sample. Powder samples were measured using a Finnigan-MAT 253 mass spectrometer coupled with a Kiel Carbonate Device at Nanjing Normal University, China. Results are reported as "delta" values, where $\delta^{18}O = \frac{R_{sample}}{R_{standard}} - 1$ relative to standard Vienna Pee Dee Belemnite (VPDB), and given in per mil (‰) notation (where $R$ is the ratio of the heavy isotope to light isotope for a material, e.g., $R = \frac{^{18}O}{^{16}O}$). Replicate analyses of an international standard (NBS19) indicated long-term reproducibility, with precisions better than 0.06‰ for δ¹⁸O.

Five ~50 mg powdered samples were drilled for triple oxygen measurements at the IsoPaleoLab, University of Michigan. Samples were acidified and converted to $O_2$ for analysis on a Nu Perspective mass spectrometer following methods[63]). Isotope data were corrected with a session specific-normalization and normalized to the VSMOW-SLAP scale[64]. The $\Delta^{47}O$ values were secondarily normalized to the Wostbrock et al.[65] IAEA-603 value of -100 peg meg VSMOW-SLAP via analyses of the equivalent IAEA-603 and IAEA-C1 standard to account for fractionations occurring in acid digestion and reduction steps of sample processing. The $\Delta^{47}O$ value is defined using "delta-prime" notation as: $\Delta^{47}O = \delta^{47}O - 0.528 \times \delta^{48}O$, given in per meg, where $\delta'^{x}O = \ln\left(\frac{R_{sample}}{R_{standard}}\right)$ (e.g., ref. 66).

## Data availability

The $^{230}Th$ dates for four stalagmites used in this study are available in Supplementary Table 1. Oxygen isotope data ($\delta^{18}O$ and $\Delta^{47}O$) for four stalagmites have been deposited in a public repository of Figshare [https://doi.org/10.6084/m9.figshare.27924426.v4][67]. Source data are provided with this paper.

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

## Acknowledgements

We thank Prof. Benjamin H. Passey for providing triple oxygen isotope measurements at the IsoPaleoLab, University of Michigan. We thank Jingwei Zhang from Jiangxi Normal University and Zhenjun Wang from Anhui Normal University for their help during the fieldwork. This research was funded by the National Natural Science Foundation of China (41931178 to (W.Y.), 42071105 to (Z.K.), 42207505 (L.Y.) and 42072207 (C.S.)).

## Author contributions

L.Y., Z.K. and W.Y. conceptualized this study. Z.K. and W.Y. supervised the work. L.Y., Z.K., W.Y. and C.S. obtained funding. Z.K. and Z.Z. provided geological samples. L.Y., H.T.E., Z.B., W.Q., S.Q., C.H. and R.L.E. performed experiments and obtained data. L.Y., H.T.E. and Z.B. performed formal analysis and visualization. L.Y. drafted the manuscript. L.Y., Z.K., W.Y., C.S., H.T.E., Z.B., W.Q., Z.Z., S.Q., C.H. and R.L.E. revised the manuscript and contributed to the manuscript.

## Competing interests

The authors declare no competing interests.
