## [Transparent Peer Review file · Nature Communications]

Asian summer monsoon variability across Termination II and implications for ice age terminations

Corresponding Author: Dr Kan Zhao

Version 0:

Reviewer comments:

Reviewer #1

(Remarks to the Author)

The manuscript titled "East Asian summer monsoon variabilities across the penultimate deglaciation and implications for ice age terminations" by Liang et al. focuses on Termination II (TII) and its differences from Termination I (TI) within the Late Quaternary glacial cycles. The authors argue that a new East Asian summer monsoon (EASM) record from Shima Cave in southern China shows a weak monsoon event at approximately 139,000 years ago, preceding TII. They suggest that this precondition may have influenced various paleoclimatic processes in the Southern Hemisphere and eventually triggered TII. However, I remain unconvinced that the 139,000-year event has global paleoclimatic implications. Furthermore, I find the correlations the authors draw among North Atlantic sea surface temperatures (SST), dust flux, freshwater flux, ice-rafted debris (IRD), Antarctic methane (CH₄), carbon dioxide (CO₂) levels, deltaD, and dust flux to be overstated, given the resolution and values involved. Therefore, I cannot recommend acceptance of this manuscript for publication in Nature Communications. Detailed suggestions for improvement are provided below:

Line 11 to 22: Please review the English grammar and ensure uniform decimal points for ages. It appears to me that the current version of the abstract lacks emphasis on the importance and novelty of this study.

Line 56 to 59: This statement may suggest bias, despite ongoing debates regarding the similarities and differences between TI and TII, as outlined by the authors above.

Line 81 to 82: Authors should correctly refer to "Mod Age" as "MOD-AGE," aligning with its original reference..

Line 90 to 96: Speleothem $\delta^{18}\text{O}$ records should be assessed for isotopically equilibrium precipitation, despite the absence of absolute methods for verification. However, the millennial-scale oscillations in the SM record examined here cannot undergo such testing due to the discontinuous dataset of the SM record itself and substantial discrepancies between various Chinese cave records presented in this manuscript (Figure 2). This limitation could significantly undermine the authors' arguments.

Figure 1: The age models of these stalagmites reveal several age outliers and reversed dates based on their stratigraphic order (Figure S2). Additionally, while Figure 1 and Table S1 present 9 ages for SM16, the MOD-AGE model in Figure S2 shows only 8 ages. To accurately estimate petrography and identify possible hiatuses, section images of the SM stalagmites are essential.

Line 105 to 106: Reference 26 is not suitable for this sentence, as the SM record in this study has not been replicated on a millennial timescale as suggested by Dorale and Liu (2009).

Line 133 to 135: Authors should provide more detailed information on the statistical analysis in the Methods section, particularly regarding the merging of data from different datasets

Line 135 to 138: I am not convinced by this interpretation because XBL and JJ caves are located slightly upstream compared to other cave sites, suggesting that these sites may reflect signals of the Indian Summer Monsoon (ISM) or a combination of ISM and East Asian Summer Monsoon (EASM).

Line 174 to 177: I am not convinced by such a detailed correlation presented by the authors because individual cave records exhibit significantly different changing patterns over time.

Line 179 to 183: These assertions seem unreasonable due to the millennial-scale differences among Chinese records and the potential data noise levels (also, sampling resolution) of SST and CH4 records.

Line 206 to 207: Please revise the figure citation to 'Figure 3, left panel' to conform to the manuscript format of Nature Communications.

Line 205 to 261: The last section titled 'Different routes to ice age terminations' is overly speculative, as the authors' main arguments lack direct support from the SM records.

Line 206 to 209: I remain unconvinced by the author's argument, as mentioned previously.

Line 245 to 247: This explanation does not align with the results presented by the authors.

Line 250 to 252: I find the authors' estimation of CO2 rise unconvincing, particularly due to the slightly higher value at ~142 ka BP compared to 140 ka BP.

Line 269 to 270: Instead of "parts mil," you should use "per mil" or "‰" to correctly denote the unit.

Reviewer #2

(Remarks to the Author)

Liang et al use 4 speleothem records from a cave in China to examine H11 and Termination II in the EASM region. They examine how the EASM record fits into the sequence of global events to explore drivers and feedback mechanisms. I think the records themselves are extremely valuable and supported with elaborate discussion.

I have a few comments that will hopefully improve the manuscript:

- Since the premise of the paper really seems to be examining the sequence of events, it would be great to see the speleothem age-model uncertainties clearly in the plots and in the text. For example, Figure 1 could include age-model uncertainties as shading, and could show expanded plots of significant time periods e.g. H11. Similarly, the text, should include the full ages and uncertainties in brackets e.g. Lines 93 to 95. Similarly for Figures 2 and 3, it would be really useful to see some consideration of temporal uncertainties on the remaining records of ice melt, CO2, temperature etc as well. You could try something like Fig. 7 in this publication: <https://cp.copernicus.org/preprints/cp-2024-37/cp-2024-37.pdf>

Line 52: What does 'ocean-forcing sea level rise' mean

Lines 121 to 125: Does T1 show this N-S pattern as well?

Line 177: Positive biases near 133 and 130 ka BP in the list of caves are not immediately apparent in the figure. Perhaps show these with lines / arrows.

Lines 189 and 190: Is the Figure labelling (Figure 3C) correct? I would also reconsider this, I find it hard to see these positive anomalies in the record.

Line 223: Should be 4000 years rather than 4 ka.

Line 236: What does a bias in $\delta^{13}\text{C}$ in the Antarctic ice core mean? Is this positive / negative? What does it indicate?

Line 257: Should be 7000 years rather than 7 ka and 3500 years rather than 3.5 ka.

Version 1:

Reviewer comments:

Reviewer #1

(Remarks to the Author)

Liang et al. have extensively revised their original manuscript in response to the reviewers' comments. I am generally satisfied with the improvements in the authors' explanations, which now focus more on their dataset. However, I would like to request further clarification on the detailed correlations of H12 in Figs. 2 and 4, as well as the pause at the end of TII.

In the abstract, you mention that the initial rise in CO2 was one of the key preconditions that led to the end of the glacial period in TII. Could you explain why this is the case?

What specific evidence supports the proposed causes of the different preconditions between T1 and TII, as outlined in the

revised manuscript?

Reviewer #3

(Remarks to the Author)

I was brought in for the second round of reviews, so I'll limit my discussion to what I deem as appropriate considering the edits which were made after the first round. In general this is a well written article, and I support publication of it in Nature Communications. However there are some issues that need to be addressed before I can sign off on publication. I will list them now in order of importance.

1) Given the errors in the age models, and the distribution of the U-Th dates relative to the d18O features, too much emphasis has been placed on "the pause" in the manuscript. While it *may be a real feature of the TII, the age constraints in these records have too much uncertainty to say for sure that it did indeed occur. There are 4 records that span 'the pause', but only one of them was used as the primary chronology: SB25. Given the time constraints among the records of the before ~ 128 ka, I can see why the authors choose this particular record. However two other records from the same cave show a large d18O decrease occurring 1000 years before SB25. While the record from Dongge agrees with the timing as SB25, the uncertainty on the U-Th ages in Dongge are large. Furthermore, there is a considerable slowdown in growth rate and possible signs of a hiatus in SB25 after 400 mm/before 129 ka (Cheng et al., Science, 2009: Figure S2, Table S1) – which is exactly when 'the pause' occurs. In fact, there is a clear degradation in the temporal resolution during this timeperiod in the main figures 2-4. Alternate plausible age models using the exact same U-Th ages would have NO pause and either a hiatus during this time - or even a more gentle change in d18O. Given the uncertainty in the age models, much of the discussions surrounding "the pause" is not supported and is conjecture at best. It should be only a small mention with large caveats about the uncertainty.

2) I'm not sure what the authors mean by 'preparing' and 'quasi-equilibrium' stages in Figure 4. This entire argument seems very odd, and I'm not sure where it's coming from or why the nomenclature is chosen. If it is based on the CO2 curves, they look almost identical to me. The change point algorithm (Figure 7 in SI) shows that they contemporaneously within error. The older CO2 record just looks like a smoothed moving average of the higher resolution younger record (most likely due to compaction of the ice), and I see no real functional difference between them. It can't be based on insulation as the authors even state that insulation during those two times is very similar. The dust records are also similar during these 'stages'. Therefore I'm not sure why these two terms are being used. They are not clearly defined, and I advocate to drop those two terms from the manuscript.

3) It would be much more illustrative to depict figures 3 and 4 as kilo years since glacial maximum (or time since glacial maximum or years since glacial maximum, e.g TSGM). They would both then start 0 and count upwards, such that one could then easily compare changes between the two records without having to do the math in their head. By subtracting off the time period since the glacial maximum (either last or penultimate) this would help to compare the timing of events in the records between CO2 meltwater pulses Heinrich Asia monsoon etc during deglaciation events. Also, starting TI at 23 ka and not 24 ka cuts off H2 from Figure 4. Then both deglaciations would have two Heinrich events in the figure. Seems arbitrary to start at 23 and have that white space on the right of the plot. The label on top of LGM in the rectangle shows the state of land ice/sea level from LR04.

4) It would be very informative for the reader to label the arrows/lines in figure 5 with some sort of nomenclature to say exactly which of the time series are being used to come up with those bullets. Currently, Figure 5 reads much more like a story rather than a logical argument. By tying these arrow bulleted points to actual points in the time series in Figures 2-4, the reader could follow along the mechanism with quantitative time series. This was partially done in the right panel of figure 6 of the supplementary information by labeling the different events on the second time scale on the right, such that one can partially follow along with the color-coded event and the color-coded time series. However, it is a little difficult to go back and forth between them just based on the color coding, and some sort of other indicator would be more helpful for the reader to follow along through the mechanism that the authors are proposing in Figure 5 of the main manuscript.

5) Some of the arrows in the figures are misleading, and I see no reason as to why they're there. For example, the two gray arrows pointing downwards on the right panel of F&G in Figure 4. The Gray arrow in the curve during the LGM at approximately 17.5 ka. This centennial scale oscillation in the record looks much like many other ones and there's no reason to call that particular one out other than the authors subjective selection. I see no quantifiable changes in the record to support it unlike the record from PGM where at 136 ka there a change in the average values during that step down phase showing a clear change in the record. Only use arrows when the math says they should be there - not to try and convince the reader to 'see' something.

Version 2:

Reviewer comments:

Reviewer #1

(Remarks to the Author)

I generally agree with the authors' additional responses, but I think there are too much speculations based on the subjective comparisons between the data presented by the authors and others. I think the authors' claims through these comparisons are not appropriate to be treated as important content in this paper, and if this part is greatly reduced and the expression is refined, I believe it can be published in Nature Communications.

Reviewer #3

(Remarks to the Author)

In general, they did a good job at responding to the reviews. Nice update on the figures. I appreciate it that they removed all mention of “the pause”, as the uncertainties in the records did not support that argument. Along the same line of logic, two of the dotted lines in Figure 2 are also not supported by uncertainties in the age models. They are cherry picked for sure. One would be hard pressed to get an objective scientist not associated with the work to look at the time series and draw those connected lines. I am referring to lines 146-148 in the revised manuscript: “Two moderate excursions in the SST record could also correlate to millennial-scale moderate monsoon shifts at approximately 137.5 ± 0.4 and 134 ± 0.6 ka BP.” I recommend that the lines be removed, but the authors are free to say this in text form – although it is not supported by the data.

I see where the authors are going with the whole ‘tipping point’ theory, as a gradual increase in insolation and CO₂ cannot explain the abrupt changes during the deglaciation. There are lots of missing dominos in the deglaciation not included in this manuscript, but it is not a comprehensive review. One wording to change would be ‘element’ in line 169. Technically, CO₂ is referred to as a forcing, not an element, with regards to climate. Please update the text accordingly.

Other than these small changes, the manuscript is ready for print.

Reviewer #1 (Remarks to the Author):

The manuscript titled "East Asian summer monsoon variabilities across the penultimate deglaciation and implications for ice age terminations" by Liang et al. focuses on Termination II (TII) and its differences from Termination I (TI) within the Late Quaternary glacial cycles. The authors argue that a new East Asian summer monsoon (EASM) record from Shima Cave in southern China shows a weak monsoon event at approximately 139,000 years ago, preceding TII. They suggest that this precondition may have influenced various paleoclimatic processes in the Southern Hemisphere and eventually triggered TII. However, I remain unconvinced that the 139,000-year event has global paleoclimatic implications. Furthermore, I find the correlations the authors draw among North Atlantic sea surface temperatures (SST), dust flux, freshwater flux, ice-rafted debris (IRD), Antarctic methane (CH₄), carbon dioxide (CO₂) levels, deltaD, and dust flux to be overstated, given the resolution and values involved. Therefore, I cannot recommend acceptance of this manuscript for publication in Nature Communications. Detailed suggestions for improvement are provided below:

Response: Thank you so much for your nice comments and suggestions which help to improve the manuscript. After reading through relevant research, we found that the 139-ka event should be related to Heinrich event 12 as pointed out by Lisiecki and Stern (2016). They also mentioned that the H12 event has a global impact. We then re-evaluated this event in the context of many other different ocean and terrestrial archives, as provided in the figures in the supplementary materials and main manuscript, and reconsidered the correlations between these records. Besides, we removed the discussion of SST and monsoon relationship during the Last Interglacial, which might be overstated.

Reference: Lisiecki, L. E. & Stern, J. V. Regional and global benthic $\delta^{18}\text{O}$ stacks for the last glacial cycle. *Paleoceanography* 31, 1368–1394 (2016).

The following part is the point-by-point response.

Line 11 to 22: Please review the English grammar and ensure uniform decimal points for ages. It appears to me that the current version of the abstract lacks emphasis on the importance and novelty of this study.

Response: Thank you so much for your nice suggestion. We reviewed the grammar thoroughly and uniformed the decimal points for ages. The abstract was also revised to highlight the importance and novelty. Please see in lines 20–29.

The detailed anatomy of Termination I (TI) is well depicted, but to which extent TI represents previous terminations remains controversial. Based on an analogous two-step structure between TI and TII observed in stalagmite-based Asian monsoon records from Shima and nearby caves in China, we found that a major CO₂ rise phase (135.7±1–129 ka BP) during TII was similar to that during TI, including rapid shifts and amplitudes. Correlating marine and ice-core records to our U/Th-dated record, we determined an initial 15–20 ppm rise of CO₂ occurring at 139±1 ka BP before major rise phase. This initial rise of CO₂, distinct from CO₂ behavior before TI, likely resulted from a 50% reduction in southern hemisphere dust flux and millennial-scale processes in the ocean and Asian monsoon. We show that the initial rise of CO₂ was one of the key preconditions that ended up glacial, supporting the “tipping point” theory.

Line 56 to 59: This statement may suggest bias, despite ongoing debates regarding the similarities and differences between TI and TII, as outlined by the authors above.

Response: Thank you so much for your nice suggestion. We modified the introduction after considering your suggestions. Please see in lines 45–62.

Current understanding of ocean-atmosphere-cryosphere interactions during terminations mainly relies on the assessment of Termination I (TI, ~18–11.7 thousand years before the present, hereafter ka BP) due to the abundance of archives with reliable, independent age constraints⁵. Denton et al.⁵ suggested that the collapse of northern hemisphere ice sheets, due to rising insolation, disrupted the global oceanic and atmospheric circulations, leading to CO₂ release from the Southern Ocean that further augmented global warming during the last termination. However, TI may not be representative of previous terminations. An increasing body of evidence from high-resolution ocean and terrestrial records suggests climatic dissimilarities between TI and TII, where TII is the penultimate ice age termination (TII, ~136–129 ka BP), containing the transition from the Penultimate Glacial Maximum (PGM, ~140 ka BP) to the Last Interglacial (LIG, ~129–116 ka BP)^{6,10–18}. No clear evidence of a Younger Dryas-like event in the North Atlantic sea surface temperature (SST)¹⁴ nor a Bølling/Allerød-like event in the AMOC intensity occurred during TII⁶. Importantly, Cheng et al.⁹ observed that monsoon structure during TII was different from TI, lacking the Bølling-Allerød period. Another notable discrepancy between TI and TII was that atmospheric CO₂ at the end of TII

was higher than at the end of TI by >15 ppm¹⁹. Accordingly, Barker and Knorr⁸ suggested that Denton et al.'s model⁵ could not explain the anatomy of all ice age terminations. To better understand why the two most recent terminations exhibited different characteristics, especially the rise in atmospheric CO₂ which plays a critical role in regulating the Earth's temperature, we must investigate the cascade of climatic elements and dynamics before and during the ice age terminations.

Line 81 to 82: Authors should correctly refer to "Mod Age" as "MOD-AGE," aligning with its original reference..

Response: Thank you so much for your nice suggestion. Changed accordingly. Please see in line 83.

The age models for Shima samples were obtained using the MOD_AGE model²¹ ...

Line 90 to 96: Speleothem δ¹⁸O records should be assessed for isotopically equilibrium precipitation, despite the absence of absolute methods for verification. However, the millennial-scale oscillations in the SM record examined here cannot undergo such testing due to the discontinuous dataset of the SM record itself and substantial discrepancies between various Chinese cave records presented in this manuscript (Figure 2). This limitation could significantly undermine the authors' arguments.

Response: Thank you so much for your nice suggestion. Previously, while we found a general changing pattern of the monsoonal deglaciation in different cave records (that is, changing from more positive values to more negative values in δ¹⁸O data), we also tried to say something about the regional discrepancies, which might be confusing. In the revised version, regional discrepancies are not discussed anymore and therefore we removed several calcite records. Considering your suggestion, we did three tests in order to test the isotopic equilibrium precipitation of speleothem δ¹⁸O, including Hendy test, Replication test and triple oxygen isotope analyses. Please see details in lines 106–128.

Stalagmite precipitation in isotopic equilibrium is a prerequisite for using its δ¹⁸O as a climate proxy. Three lines of evidence suggest that Shima δ¹⁸O records reflect formation in isotopic equilibrium. First, the Hendy test²⁷ was performed on four individual growth layers of each sample. Most δ¹⁸O and δ¹³C variations along the same layer range from 0.1‰ to 0.3‰ (Fig. 1b), and the absence of positive relationships between δ¹⁸O and δ¹³C are consistent with the equilibrium formation of calcite (Supplementary Fig. 3). Second, replication of different stalagmite δ¹⁸O records is a reliable method to test for equilibrium fractionation in carbonates²⁸. Our Shima records display good consistency with other cave records from southern China on the orbital to millennial timescales, including abrupt positive shifts of ~2‰ at approximately 136 ka BP and negative shifts of >4‰ from the TII to LIG (Supplementary Fig. 4). Third, the Δ¹⁷O vs. δ¹⁸O trend of Shima stalagmite SM7 has a slope of ≈ 1 per meg/‰ (Fig. 1c), consistent with the predominance of rainout processes in determining δ¹⁸O²³. Similar trends could mathematically come from combinations of multiple processes. However, even combining trends from, for example, rainout (Rayleigh, 0 per meg/‰) and within-cave kinetic (7 per meg/‰) processes would still require an overwhelming 86% of the signal to be coming from climate-related drivers (i.e., 0.86×0 per meg/‰ + 0.14×7 per meg/‰ = 1 per meg/‰). Changes in Δ¹⁷O can also come from varying the relative humidity during evaporation at the moisture source (-9.5 per meg/‰) and require only a 37.5 % contribution from the climate signal (i.e., 0.375×(-9 per meg/‰) + 0.625×7 per meg/‰ = 1 per meg/‰). These calculations, in combination with the fact that modern precipitation in the Asian monsoon region shows weak, slightly negative to slightly positive trends (Supplementary Fig. 5), also support stalagmite formation in (near-)isotopic equilibrium. Ultimately, while these lines of evidence are useful in establishing that the Shima samples formed in (near-)isotope equilibrium, our fundamental interpretations below are about the timing of change, not the exact interpretation of speleothem δ¹⁸O, and thus independent of the driving processes.

Figure 1: The age models of these stalagmites reveal several age outliers and reversed dates based on their stratigraphic order (Figure S2). Additionally, while Figure 1 and Table S1 present 9 ages for SM16, the MOD-AGE model in Figure S2 shows only 8 ages. To accurately estimate petrography and identify possible hiatuses, section images of the SM stalagmites are essential.

Response: Thank you so much for your nice suggestion. Yes, there are some reversed dates in samples and therefore we chose to use modeling for the chronology reconstruction. We checked the dates and age models for these samples again. There are 8 dates for SM16. Besides, we put the section images for these samples in Supplementary Fig. 2 as you suggested. Short-lived hiatuses might exist, but considering dating errors for several centuries, the exact timing for hiatuses is hard to detect. Nonetheless, modeled chronologies run through most ²³⁰Th dates.

Line 105 to 106: Reference 26 is not suitable for this sentence, as the SM record in this study has not been replicated on a millennial timescale as suggested by Dorale and Liu (2009).

Response: Thank you so much for your suggestion. The replication test is important for stalagmite studies, and selecting proper records is necessary before comparison. We removed records from northern China because moisture sources and atmospheric circulation systems for northern China are different from those for southern China (Chu et al., 2019; Zhang et al., 2021), which could cause different isotopic signals in rainfall and speleothem. We also removed Xiaobailong and Jiangjun records which have very low resolutions and fewer dating constraints. All the records from southern China presented show an overall similarity, including the 139-ka and 136-ka weak monsoon intervals and a rapid monsoon intensification around 129 ka BP (Supplementary Fig. 4 Supplementary Fig. 4). Besides, to alleviate the temporal gap of the abrupt Asian monsoon TII in Shima record, we combined our records with that of Sanbao Cave (270 km distant), which is well-dated and has dating errors of less than 100 years by adding a systematic bias of -0.2‰ to SB25 $\delta^{18}\text{O}$ data.

References:

Chu Q et al. Roles of moisture sources and transport in precipitation variabilities during boreal summer over East China. *Clim Dyn* 53, 5437–5457 (2019).

Zhang R et al. Summertime Moisture Sources and Transportation Pathways for China and Associated Atmospheric Circulation Patterns. *Front. Earth Sci.* 9:756943 (2021).

Line 133 to 135: Authors should provide more detailed information on the statistical analysis in the Methods section, particularly regarding the merging of data from different datasets

Response: Thank you so much for your nice suggestion. We deleted the merging data from different datasets which is not important in the present version.

Line 135 to 138: I am not convinced by this interpretation because XBL and JJ caves are located slightly upstream compared to other cave sites, suggesting that these sites may reflect signals of the Indian Summer Monsoon (ISM) or a combination of ISM and East Asian Summer Monsoon (EASM).

Response: Thank you so much for your nice suggestion. According to published papers, the XBL and JJ caves should reflect more of the signals of the Indian Summer Monsoon (ISM) or a combination of ISM and East Asian Summer Monsoon (EASM) (Wassenburg et al., 2020; Cai et al., 2015). In the revised version, we use the term “Asian summer monsoon”, which includes both systems of the ISM and EASM for discussion. This interpretation is generally accepted in previous studies (e.g. Cheng et al., 2016; Kelly et al., 2006; Wang et al., 2008).

Reference:

Wassenburg, J. A. et al. Penultimate deglaciation Asian monsoon response to North Atlantic circulation collapse. *Nat. Geosci.* 14, 937–941 (2021).

Cai, Y. et al. Variability of stalagmite-inferred Indian monsoon precipitation over the past 252,000 y. *Proc. Natl. Acad. Sci. U.S.A.* 112, 2954–2959 (2015).

Wang, Y. et al. Millennial- and orbital-scale changes in the East Asian monsoon over the past 224,000 years. *Nature* 451, 1090–1093 (2008).

Cheng, H. et al. The Asian monsoon over the past 640,000 years and ice age terminations. *Nature* 534, 640–646 (2016).

Kelly, M. J. et al. High resolution characterization of the Asian Monsoon between 146,000 and 99,000 years B.P. from Dongge Cave, China and global correlation of events surrounding Termination II. *Palaeogeogr. Palaeoclimatol. Palaeoecol.* 236, 20–38 (2006).

Line 174 to 177: I am not convinced by such a detailed correlation presented by the authors because individual cave records exhibit significantly different changing patterns over time.

Response: Thank you so much for your nice suggestion. We deleted this expression.

Line 179 to 183: These assertions seem unreasonable due to the millennial-scale differences among Chinese records and the potential data noise levels (also, sampling resolution) of SST and CH₄ records.

Response: Thank you so much for your nice suggestion. We modified this part and deleted some overstated expression. The link between 134-ka CH₄ peak and a short-lived strengthening of the Asian summer monsoon might be correct because it has been discussed by Schmidely et al. (2021).

Reference:

Schmidely, L. et al. CH₄ and N₂O fluctuations during the penultimate deglaciation. *Clim. Past* 17, 1627–1643 (2021).

Line 206 to 207: Please revise the figure citation to 'Figure 3, left panel' to conform to the manuscript format of Nature Communications.

Response: Thank you so much for your nice suggestion. We modified the figure citation in the revised version thoroughly.

Line 205 to 261: The last section titled 'Different routes to ice age terminations' is overly speculative, as the authors' main arguments lack direct support from the SM records.

Response: Thank you so much for your nice suggestion. In the revised version, we highlighted the older section of Shima records and discussed more comprehensively the preconditions before termination II. Compared with other Chinese cave records, Shima record has smaller ^{230}Th age errors for this part and higher data resolutions (Supplementary Fig. 4). Based on a combined record of Sanbao and Shima, we used cave-marine and cave-ice core correlations to establish the linkages between ocean-atmosphere processes, which allowed us to discuss the main points regarding the 139 ka event. Although we did not change the title, we modified this section to be more readable. Please see in lines 206-263.

Different routes to Ice Age terminations

While TI and TII have similarities in their overall rate of CO_2 change and the presence of millennial-scale events, they have distinct patterns before the onset of the terminations. During the "preparing stage" of the PGM, CO_2 had already increased gradually from 139 ± 1 ka BP to 135.7 ± 1 ka BP, leading to a total rise of 15–20 ppm before the abrupt increase (Fig. 3a). However, CO_2 concentration was stable at 190 ppm for about 4000 years prior to the onset of TI. Orbital forcing and global ice volume during the LGM (Fig. 4A, B) were similar during the PGM (Fig. 4a, b), both of which provide critical backgrounds for CO_2 variations during glacial-interglacial cycles⁴², and therefore cannot explain differences in CO_2 variations between terminations.

We notice three significantly different conditions during PGM and LGM: (i) a more unstable monsoon status during the PGM than LGM, (ii) the meltwater pulse (MWP) event 2A¹¹ associated with a Heinrich event (H12) occurring at ~ 139 ka BP⁴³ and (iii) a significant drop in dust flux recorded in the Antarctic ice cores⁴⁴. During the "quasi-equilibrium" LGM⁴⁵, the high-resolution Chinese composite record²⁵ displays an increasing trend in monsoon intensity with small amplitude and multi-centennial-scale changes forced by moderate variations in the AMOC⁴⁵ (Fig. 3C). However, during the PGM, a significant 139 ± 0.6 -ka WMI is detected in addition to centennial oscillations. This WMI was likely tele-connected with the H12 event and the MWP-2A (Fig. 4d, e), as well as the North Atlantic SST cooling^{3,11} (Supplementary Fig. 6). Although not as intensive as MWP-2B which contributed $\sim 70\%$ of deglacial sea level rise¹¹, MWP-2A was the initial ice-sheet retreat before TII and contributed about 30 meters sea level rise^{46,47}, which may have disturbed Southern Ocean stratification and lead to deep ocean CO_2 release.

More importantly, the 50% off in dust flux observed at ~ 139 ka BP (Fig. 4f) could have also contributed to CO_2 rise and ultimately arose from less exposure of the South American continent due to sea level rise (Fig. 4b) and a relatively humid South America. On millennial timescales, weakening of the ASM is generally coincident with strengthening of the South American summer monsoon (SASM)⁴⁸, and South America, especially Patagonia, is the primary dust source for Antarctica⁴⁹. At around 139 ka BP, the decrease in dust input to Antarctica and the Southern Ocean might be due to relatively humid conditions⁴⁸ and the well-developed vegetation⁵⁰ in South America, both associated with a relatively strong SASM which contradicts the ASM intensity. Increased input of Fe-bearing atmospheric dust to the Fe-deficient Southern Ocean may have stimulated the primary productivity of phytoplankton and enhanced oceanic sequestration of CO_2 during glacial periods, in contrast to changes observed during interglacial periods⁵¹. It has been estimated that ~ 40 ppm of the change in CO_2 concentration during the glacial-interglacial transitions is caused by changing dust export to the Southern Ocean⁵². Nearly synchronous changes in Antarctic dust flux and atmospheric CO_2 from $\sim 140\pm 1$ to 135.7 ± 1 ka BP suggest that changes in biological pump and net primary productivity in the Southern Ocean were tightly coupled and already began before the onset of TII.

The 15–20 ppm CO_2 increase before 135.7 ± 1 ka BP might explain why atmospheric CO_2 concentration at the end of TII is >15 ppm higher than the Holocene. This initial, gradual rise of CO_2 can be regarded, according to the "tipping point" theory³⁶, as an early warning signal that leads to the inevitable CO_2 rise at the onset of TII. From the viewpoint of physical mechanisms, the abrupt rise of CO_2 since 135.7 ± 1 ka BP was likely due to the increased carbon storage in the stratified Southern Ocean during the preceding PGM⁵³, and thus the early warning signal for TII arose from the southern hemisphere. In contrast, the early warning signal for TI was particularly expressed in the

northern hemisphere^{36,54,55}. Brovkin et al.³⁶ suggested a tipping point at 18 ka BP when AMOC collapsed, before which a repeated build-up of a subsurface heat reservoir in the subpolar Atlantic was robust⁵⁴. That the early warning signal may occur in either hemisphere suggests that a slow, initial process, regardless of location, is a precondition for shifting from glacial to interglacial conditions.

In Denton et al.'s model⁵, higher CO₂ driven by the reorganization of ocean and atmospheric circulation is a key factor in completing the termination. Here our hypothesis further points to its importance in initiating the termination (Fig. 5). Under similar orbital configurations of large ice sheets and low northern hemisphere solar insolation (NHSI), the PGM experienced more dramatic climate oscillations. The H12 and the MWP-2A events caused a reduction in ASM and ~30m sea level rise^{46,47}, leading to an abrupt 50% decrease in southern hemisphere dust flux. All these processes led to the initial rise in CO₂ that ultimately gave feedback on the extensive ice sheets ready for the onset of TII. When gradual changes in ocean temperature or CO₂ crossed the threshold for “tipping”, forced by rising NHSI, the deglaciation process started, following the sequence of massive ice sheet melting, reduced AMOC, weakened ASM and southward Intertropical convergence zone (ITCZ)/westerlies as summarized by Denton et al.⁵. All these processes caused continuous CO₂ release from the Southern Ocean to sustain ice sheet melting, which then ended up with a last hit of the 1000-year-long “pause”.

Line 206 to 209: I remain unconvinced by the author's argument, as mentioned previously.

Response: Thank you so much for your suggestion. We re-evaluated global geological archives and found that the 139 ka event should have occurred and had a global impact because it is a Heinrich-type event (Lisiecki and Stern, 2016). Please see the modifications in lines 206-263.

Line 245 to 247: This explanation does not align with the results presented by the authors.

Response: Thank you so much for your nice suggestion. Original writing might be confusing and so we rewrote this part in the section “Different routes to Ice Age Terminations”. Please see in lines 206-263.

Line 250 to 252: I find the authors' estimation of CO₂ rise unconvincing, particularly due to the slightly higher value at ~142 ka BP compared to 140 ka BP.

Response: Thank you so much for your nice suggestion. We provide constraints on the timing of initial CO₂ rises by bridging the ice core, marine and cave records, as discussed in the section “Timing of the initial rise of CO₂ during the Penultimate Glacial Maximum”. From the long-term development of the termination, we recognize that the initial CO₂ rise might start from ~140 ka BP, and this timing point is supported by three CO₂ records. Please see in lines 130–159.

Timing of the initial rise of CO₂ during the Penultimate Glacial Maximum

Based on a mechanical link of abrupt shifts in ASM (under whose influence the wetland is one of the major sources for atmospheric CH₄) and CH₄ records in Antarctic ice cores during ice age termination^{9,29}, we compare the Shima-Sanbao record with CH₄ and CO₂ records from the EDC ice core on the AICC2012 chronology^{30,31}. Apart from the nearly-consistent abrupt intensification/rises in ASM and CH₄ at around 129 ka BP (blue dashed line Fig. 2b, c), a small peak of CH₄ at ~134 ka BP is possibly tied to a short-lived strengthening of the ASM, which has been confirmed in a previous study³² and further constrains the chronology of CO₂ and CH₄ records for older sections. All three atmospheric CO₂ records began to rise at ~140 ka BP until 129 ka BP, lasting for approximately 11000 years with a total rise of >100 ppm (Fig. 2a). The onset timing of CO₂ rise cannot be determined more precisely due to the low-resolution data of CO₂ at ~140 ka BP. However, considering that: (i) the linkage of North Atlantic ice-rafted debris (IRD) events or H stadials with rises of CO₂ through a mechanism of the AMOC weakening that promotes the release of CO₂ from the Southern Ocean to the atmosphere⁵ and (ii) good correlations of WMIs and IRD events^{20,25}, here we can provide constraints on the timing of initial CO₂ rises by bridging the ice core, marine and composite Shima-Sanbao cave records.

There is a broad similarity between our cave record and a North Atlantic SST record³³, including dramatic monsoon weakening and strengthening associated with a cooling of ~4°C at the beginning of TII and a large amplitude warming at the end of TII in the North Atlantic (black dashed lines in Fig. 2). Two moderate excursions in the SST record could also correlate to, monsoon weakening and strengthening at approximately 137.5±0.4 and 134±0.6 ka BP. Since the chronologies for ODP984 and MD01-2444 cores were well aligned by tie points³³, major

IRD peaks in core ODP984³⁴ can also be well linked to the sequence of WMIs in the cave record, including H12 and H11. Other IRD records on independent age models also show clear peaks and support the tight correlations with WMIs (Supplementary Fig. 6). Based on the evidence presented, we propose that H12, aligning to a WMI event dated at 139 ± 0.6 ka BP, possibly drove the initial rise of CO₂ through the weakening or shutdown of the AMOC^{5,9} (Supplementary Fig. 6). The timing of WMI during the PGM in the Shima record is consistent with other independently dated Chinese cave records within their dating errors (blue arrows in Supplementary Fig. 4). By evaluating ²³⁰Th age control for cave records (Supplementary Fig. 4) and the original age model for ice cores^{30,31}, we suggest that the initial rise of CO₂ for TII occurred at 139 ± 1 ka BP.

Line 269 to 270: Instead of "parts mil," you should use "per mil" or "‰" to correctly denote the unit.
Response: Thank you so much for your nice suggestion. Changed accordingly. Please see in line 273.
.... and given in per mil (‰)....

Reviewer #2 (Remarks to the Author):

Liang et al use 4 speleothem records from a cave in China to examine H11 and Termination II in the EASM region. They examine how the EASM record fits into the sequence of global events to explore drivers and feedback mechanisms. I think the records themselves are extremely valuable and supported with elaborate discussion. I have a few comments that will hopefully improve the manuscript:

Response: Thank you so much for your comments which help to improve the manuscript. We modified the content point-by-point as follows.

- Since the premise of the paper really seems to be examining the sequence of events, it would be great to see the speleothem age-model uncertainties clearly in the plots and in the text. For example, Figure 1 could include age-model uncertainties as shading, and could show expanded plots of significant time periods e.g. H11. Similarly, the text, should include the full ages and uncertainties in brackets e.g. Lines 93 to 95. Similarly for Figures 2 and 3, it would be really useful to see some consideration of temporal uncertainties on the remaining records of ice melt, CO₂, temperature etc as well. You could try something like Fig. 7 in this publication:

<https://cp.copernicus.org/preprints/cp-2024-37/cp-2024-37.pdf>

Response: Thank you so much for your nice comments. According to your suggestion, we plot Fig. 1a with line shading of age-model uncertainties. Because our record does not cover a complete H11, we do not show an expanded plot. Besides, we add descriptions of age uncertainties in the revised version throughout the paper in records of stalagmite and CO₂. Age uncertainties in other cave records are shown in Supplementary Fig. 4, which was part of the previous Fig. 2. Uncertainties for ice core and marine records are as large as ~1500 to 3000 years across penultimate termination according to references (Mokeddem and McManus, 2016; Irvallı et al., 2016; Böhm et al., 2015; Lisiecki and Stern, 2016; Bazin et al., 2013; Tzedakis et al., 2018; Martrat et al., 2014; Marino et al., 2015; Deaney et al., 2017), and some marine records were tuned to AICC2012 chronology (Deaney et al., 2017). Most of these age models have been well discussed, verified and tested (some of them have been compared with stalagmite-based chronologies), and thus we think it is proper to use records on their independent chronologies. We also presented a temporal uncertainty plot and added it to Supplementary Fig. 6. Because different records for SST, AMOC and IRD inputs have different uncertainties, we here apply the smallest age errors with the best age constraints provided by references in Supplementary Fig. 6.

Besides, the age errors for CO₂ are constrained by U/Th-dated cave records and narrowed to 1000 years because (i) Asian monsoon (under whose influence the wetland is one of the major sources for atmospheric CH₄) can be mechanically linked to atmospheric CH₄ as shown in Fig. 1, and (ii) the chronology for EDC CH₄ and CO₂ records are all on the AICC2012 chronology (Bazin et al., 2013). An estimation of 1000 years approximate to cave dates Supplementary Fig. 4) is used for temporal uncertainties on CO₂.

References:

- Mokeddem, Z. & McManus, J. F. Persistent climatic and oceanographic oscillations in the subpolar North Atlantic during the MIS 6 glaciation and MIS 5 interglacial. *Paleoceanography* **31**, 758–778 (2016).
- Irvallı, N. et al. Evidence for regional cooling, frontal advances, and East Greenland Ice Sheet changes during the demise of the last interglacial. *Quat. Sci. Rev.* **150**, 184–199 (2016).
- Lisiecki, L. E. & Stern, J. V. Regional and global benthic $\delta^{18}\text{O}$ stacks for the last glacial cycle. *Paleoceanography* **31**, 1368–1394 (2016).
- Deaney, E. L., Barker, S. & Van De Flierdt, T. Timing and nature of AMOC recovery across Termination 2 and magnitude of deglacial CO₂ change. *Nat. Commun.* **8**, 14595 (2017).
- Böhm, E. et al. Strong and deep Atlantic meridional overturning circulation during the last glacial cycle. *Nature* **517**, 73–76 (2015).
- Martrat, B., Jimenez-Amat, P., Zahn, R. & Grimalt, J. O. Similarities and dissimilarities between the last two deglaciations and interglaciations in the North Atlantic region. *Quat. Sci. Rev.* **99**, 122–134 (2014).
- Marino, G. et al. Bipolar seesaw control on last interglacial sea level. *Nature* **522**, 197–201 (2015).
- Tzedakis, P. C. et al. Enhanced climate instability in the North Atlantic and southern Europe during the Last Interglacial. *Nat. Commun.* **9**, 4235 (2018).
- Bazin, L., et al. An optimized multi-proxy, multi-site Antarctic ice and gas orbital chronology (AICC2012): 120–800 ka, *Clim. Past*, **9**, 1715–1731 (2013).

Line 52: What does 'ocean-forcing sea level rise' mean

Response: Thank you so much for your question. It is wrongly spelled and should be “Sea level rise”. But this sentence was removed in adjustment to the Introduction.

Lines 121 to 125: Does TI show this N-S pattern as well?

Response: Thank you so much for your question. This is a good question, and we are also interested in finding such a pattern. But up till now we have not seen any published results from Shangxiaofeng Cave during TI, and we cannot say definitely whether N-S pattern existed during TI. Future work is necessary.

Line 177: Positive biases near 133 and 130 ka BP in the list of caves are not immediately apparent in the figure. Perhaps show these with lines / arrows.

Response: Thank you so much for your nice suggestion. We modified this part in the revised version and deleted the comparison of different patterns between northern and southern China. Because we want to focus more on the discussion of termination. Hope you find the revised version interesting.

Lines 189 and 190: Is the Figure labelling (Figure 3C) correct? I would also reconsider this, I find it hard to see these positive anomalies in the record.

Response: Thank you so much for your nice suggestion. In the previous version, the label should be Figure 3A. Considering your suggestion, we deleted the discussion of SST and monsoon correlation during the Last Interglacial, which might be overstated.

Line 223: Should be 4000 years rather than 4 ka.

Response: Thank you so much for your nice suggestion. Changed accordingly and we modified similar expressions in the revised version thoroughly.

Line 236: What does a bias in $\delta^{13}\text{C}$ in the Antarctic ice core mean? Is this positive / negative? What does it indicate?

Response: Thank you so much for your question. According to the study by Schneider et al. (2013), it is a ~0.1–0.4‰ decrease to lighter values of the $\delta^{13}\text{C}_{\text{atm}}$ level from ~140 000 yr BP and it can be explained by changes in global carbon storage. Generally, a shift to lighter $\delta^{13}\text{C}_{\text{atm}}$ values indicates a decline in the terrestrial biosphere. In the revised version, we removed this expression.

Line 257: Should be 7000 years rather than 7 ka and 3500 years rather than 3.5 ka.

Response: Thank you so much for your nice suggestion. Changed accordingly and we modified similar expressions in the revised version thoroughly.

Reviewer #1 (Remarks to the Author):

Liang et al. have extensively revised their original manuscript in response to the reviewers' comments. I am generally satisfied with the improvements in the authors' explanations, which now focus more on their dataset. However, I would like to request further clarification on the detailed correlations of H12 in Figs. 2 and 4, as well as the pause at the end of TII.

In the abstract, you mention that the initial rise in CO₂ was one of the key preconditions that led to the end of the glacial period in TII. Could you explain why this is the case?

What specific evidence supports the proposed causes of the different preconditions between TI and TII, as outlined in the revised manuscript?

Response: Thank you so much for your comments which help a lot in improving this manuscript. According to your suggestion, we added clarification on the detailed correlations of H12 in Figs. 2 and 4. We also deleted the overstated discussion of the pause at the end of TII.

The revised text is in lines 150-158: Since the chronologies for ODP984 and MD01-2444 cores were well aligned by tie points³⁴, major IRD peaks of H11 and H12 events in core ODP984³⁵ could also be well linked to the sequence of WMIs in the cave record (Fig. 2e). Obvious peaks for H12 event in other IRD records on independent age models varied between 140 and 138.5 ka BP and H11 event had a more complex structure with its onset at approximately 136±1.5 ka BP (Supplementary Fig. 6). These two major IRD peaks were consistent with AMOC weakening, and tightly correlated with WMIs (Supplementary Fig. 6). Based on the evidence presented, we propose that H12, aligning to a WMI event dated at 139±0.6 ka BP, possibly contributed to the initial rise of CO₂ through the weakening or shutdown of the AMOC^{5,9} (Supplementary Fig. 6e, f).

And in lines 214-223: The H12 event occurred near the boreal summer insolation minimum (Fig. 4a), and may have preconditioned TII by creating smaller, partially deglaciated ice sheets⁴². Alike the H11 event which caused an intensive reduction of the northern ice sheets and the MWP-2B¹¹, the H12 event could have caused the MWP-2A. Although not as intensive as MWP-2B which contributed ~70% of deglacial sea level rise¹¹, MWP-2A was the early phase of ice-sheet retreat before TII and contributed about 30 meters of sea level rise^{44,45}, which may have disturbed Southern Ocean stratification and lead to deep ocean CO₂ release, as well as impacting the monsoon intensities. The coincidence of the H12 event, MWP-2A, the WMI and the North Atlantic cooling episode (Figs. 2, 4) support a clear millennial-scale climatic oscillation at around 139 ka BP.

Several reasons explain why the initial rise in CO₂ during the glacial maximum was one of the key preconditions that led to the end of the glacial period in TII. Firstly, CO₂ is one of the primary greenhouse gases, and an essential amplifier of global temperature (Shakun et al., 2012). Simulations show that changes in CO₂ can contribute to 40–65% of the temperature changes during transitions between glacial and interglacial periods (Lorius et al., 1990). The initial CO₂ increase by ~15–20 ppm equals 20–25% of the total CO₂ increment during TII (80 ppm), and it might contribute to nearly 10% of the global temperature increase if we multiply by 40–65% (Lorius et al., 1990). Secondly, the amount of initial CO₂ rise is nearly equivalent to that during Heinrich events. Studies show that CO₂ increased during Heinrich events before warming in the northern hemisphere and continued rising after the onset of Dansgaard-Oeschger event (Bauska et al., 2021; Ahn and Brook, 2008). Changes in atmospheric CO₂ can lead to or may be required for Dansgaard-Oeschger variability (Menviel et al., 2020). It is also estimated that the centennial-scale feedback strength of CO₂ to global mean temperature changes is $0.155 \pm 0.035 \text{ Wm}^{-2} \text{ K}^{-1}$ during Dansgaard-Oeschger events (Liu et al., 2022). These lines of evidence indicate that even with a small amount of increase like during Heinrich events, changes in CO₂ were able to influence global or hemispheric temperature. Thirdly, using the chronological benchmark of speleothem record as described in the manuscript, the initial CO₂ rise started at 139±1 ka BP, nearly consistent with or slightly lagging the minimum of 21 June insolation at 65°N and followed the insolation rising until a rapid increase at ~135.7±1 ka BP. The 135.7±1 ka BP is also a time point when many climatic elements started the deglacial process together, including massive ice sheet collapse, the AMOC stagnation, monsoon weakening and Antarctic warming. The initial CO₂ rise thus led the onset of TII by thousands of years, within the estimated range of the lagged response of ice sheets to CO₂ change by 1000 to 4000 years (Ruddiman, 2006). Therefore, considering the feedback of CO₂ on atmospheric warming and ice

sheet melting, we suggest that ~4000–5000-year-long gradual changes in CO₂ along with insolation rise could have preconditioned ice-age terminations and contributed to the following deglaciation process.

In the manuscript, Figure 5 shows that the orbital forcing (rising from the low insolation values) and global ice volume (with δ18O values of around 5 per mil) were almost similar during the LGM and PGM, while internal climatic oscillations were different, especially the millennial-scale IRD events and southern hemisphere dust flux. We pointed out the most two important causes in the revised abstract in lines 24–27:

The major rise phases of CO₂ were comparable during TI and TII, but the initial CO₂ rise before TII was distinct from CO₂ behavior before TI, likely forced by the Earth's internal variabilities, in particular an ice-sheet collapse event and a 50% reduction in southern hemisphere dust flux.

and also, in discussion in lines 237-238:

Therefore, a combination of dust flux decreases and the H12 event could possibly cause the initial CO₂ rise, leading to different CO₂ change patterns during the LGM and PGM.

Different configurations of ice sheets might contribute to different preconditions for TI and TII, but this idea is not mature and needs to be discussed in the future because we cannot solve it now. Although the total amount of global ice volumes was nearly equivalent during both the LGM and PGM according to the LR04 record, the extent, height and size of a particular northern hemisphere ice sheet could differ. Previous work suggests a potential for a larger PGM Eurasian ice sheet, nearly triple that of the LGM Eurasian ice sheet, which means a smaller North American Ice Sheet during the PGM than LGM (Colleoni et al., 2016). However, a recent study shows a much larger PGM North American Ice Sheet than during the LGM by around 6 m sea level equivalent (Patterson et al., 2024). Different configurations of ice sheet and ice sheet changes will alter atmospheric and oceanic circulations (Zhang et al., 2014; Wang et al., 2024). It remains unclear how the ice sheet configurations were and thus additional proxy records and modeling are still needed to provide knowledge of ice sheet growth and evolutions during the PGM.

References:

- Shakun, J. D. et al. Global warming preceded by increasing carbon dioxide concentrations during the last deglaciation. *Nature* 484, 49–54 (2012).
- Lorius, C., et al. The ice-core record: climate sensitivity and future greenhouse warming. *Nature* 347, 139–145 (1990).
- Bauska, T.K., et al. Abrupt changes in the global carbon cycle during the last glacial period. *Nat. Geosci.* 14, 91–96 (2021).
- Ahn, J., & Brook, E. J. Atmospheric CO₂ and climate on millennial time scales during the last glacial period. *Science* 322, 5898, 83–85 (2008).
- Menviel, L.C., et al. An ice–climate oscillatory framework for Dansgaard–Oeschger cycles. *Nat Rev Earth Environ* 1, 677–693 (2020).
- Liu, M., et al. Past rapid warmings as a constraint on greenhouse-gas climate feedbacks. *Commun Earth Environ* 3, 196 (2022).
- Ruddiman, W. F. Ice-driven CO₂ feedback on ice volume. *Clim. Past* (2006).
- Stoll, H.M., et al. Rapid northern hemisphere ice sheet melting during the penultimate deglaciation. *Nat Commun* 13, 3819 (2022).
- Patterson, V. L., et al. (2024). Contrasting the Penultimate Glacial Maximum and the Last Glacial Maximum (140 and 21 ka) using coupled climate-ice sheet modelling. *Climate of the Past*, 20(10), 2191-2218.
- Zhang, X., et al. Abrupt glacial climate shifts controlled by ice sheet changes. *Nature* 512, 290–294 (2014).
- Wang, H., et al. Westerly and Laurentide ice sheet fluctuations during the last glacial maximum. *npj Clim Atmos Sci* 7, 213 (2024).

Reviewer #3 (Remarks to the Author):

I was brought in for the second round of reviews, so I'll limit my discussion to what I deem as appropriate considering the edits which were made after the first round. In general this is a well written article, and I support publication of it in Nature Communications. However there are some issues that need to be addressed before I can sign off on publication. I will list them now in order of importance.

1) Given the errors in the age models, and the distribution of the U-Th dates relative to the d18O features, too much emphasis has been placed on “the pause” in the manuscript. While it *may be a real feature of the TII, the age constraints in these records have too much uncertainty to say for sure that it did indeed occur. There are 4 records that span ‘the pause’, but only one of them was used as the primary chronology: SB25. Given the time constraints among the records of the before ~ 128 ka, I can see why the authors choose this particular record. However two other records from the same cave show a large d18O decrease occurring 1000 years before SB25. While the record from Dongge agrees with the timing as SB25, the uncertainty on the U-Th ages in Dongge are large. Furthermore, there is a considerable slowdown in growth rate and possible signs of a hiatus in SB25 after 400 mm/before 129 ka (Cheng et al., Science, 2009: Figure S2, Table S1) – which is exactly when ‘the pause’ occurs. In fact, there is a clear degradation in the temporal resolution during this timeperiod in the main figures 2-4. Alternate plausible age models using the exact same U-Th ages would have NO pause and either a hiatus during this time - or even a more gentle change in d18O. Given the uncertainty in the age models, much of the discussions surrounding “the pause” is not supported and is conjecture at best. It should be only a small mention with large caveats about the uncertainty.

Response: Thank you so much for your comments which help a lot in improving this manuscript. Your concern about "the pause" is worthy of consideration. Although Dongge and Hulu records can somewhat support the "pause" in Sanbao record (Supplementary Fig. 6), they have much lower resolution and larger dating uncertainties. We agree with your suggestion and deleted the discussion over the "pause" and focus more on our new data. Please see the text in lines 177-195:

After obtaining the onsets of rapid CO₂ rises for the last two ice age terminations, we established their analogy (Fig. 3). Changes in monsoon intensity inferred from cave records and Antarctic temperature inferred from ice δD records are comparable. Although the internal sequences of monsoonal events are distinguished between TI and TII⁹, analogous changes in cave records exist, including: (i) similarly abrupt weakenings and fluctuations in the ASM occurring at the transitions from glacial maximums to TI/TII (green bar in Fig. 3), and (ii) a rapid monsoon intensification after the long-term WMIs which marks an end of ice age termination (yellow bar in Fig. 3). The onset and end of TI/TII bracketed the major rise phase of CO₂, which lasted approximately 6,000 to 7,000 years. The TI and TII shared a similarly changing amplitude of ~80 ppm during the major rise phases of CO₂, regardless of different substages within them (Fig. 3b). Besides, Antarctic δD records fluctuated around -440‰ during both glacial maximums and then took similar amounts of time through deglacial processes to reach their interglacial plateaus, despite the interruption of Antarctic Cold Reversal in TI. Landais et al.³⁸ reported that atmospheric CO₂ concentrations and Antarctic temperature started increasing in phase around 136 ka BP, supporting the result of 135.7±1 ka BP here. The beginning of the rapid CO₂ rise at around 135.7±1 ka BP was also widely consistent with abrupt changes in a number of oceanic and terrestrial records like the H11 event, cooling in the North Atlantic SST and European surface temperature, as well as the AMOC weakening (Supplementary Fig. 6). Therefore, abrupt shifts in different archives at 135.7±1 ka BP could be critical changes in the global climate system, which shares similar features with the onset of TI^{10,14,39,40}.

2) I'm not sure what the authors mean by ‘preparing’ and ‘quasi-equilibrium’ stages in Figure 4. This entire argument seems very odd, and I'm not sure where it's coming from or why the nomenclature is chosen. If it is based on the CO₂ curves, they look almost identical to me. The change point algorithm (Figure 7 in SI) shows that they contemporaneously within error. The older CO₂ record just looks like a smoothed moving average of the higher resolution younger record (most likely due to compaction of the ice), and I see no real functional difference between them. It can't be based on insulation as the authors even state that insulation during those two times is very similar. The dust records are also similar during these ‘stages’. Therefore I'm not sure why these two terms are being used. They are not clearly defined, and I advocate to drop those two terms from the manuscript.

Response: Thank you so much for your comments. Sorry for the confusion in the statements. The "quasi-equilibrium" is generally used for describing the LGM because the cold climate persisted for millennia and the climate was broadly in a near-equilibrium state (Schneider von Deimling et al., 2006; Annan et al., 2022). The "preparing" stage was aimed to mention that the climate was more variable during the PGM and was preparing for the onset of TII. Considering your comments and the readability of the manuscript, we removed the terms from the text and used glacial maximums instead.

References:

Schneider von Deimling, T., et al. (2006) Climate sensitivity estimated from ensemble simulations of glacial climate, *Clim. Dynam.*, 27, 149–163.
 Annan, J. D., et al. (2022) A new global surface temperature reconstruction for the Last Glacial Maximum, *Clim. Past*, 18, 1883–1896.

3) It would be much more illustrative to depict figures 3 and 4 as kilo years since glacial maximum (or time since glacial maximum or years since glacial maximum, e.g TSGM). They would both then start 0 and count upwards, such that one could then easily compare changes between the two records without having to do the math in their head. By subtracting off the time period since the glacial maximum (either last or penultimate) this would help to compare the timing of events in the records between CO2 meltwater pulses Heinrich Asia monsoon etc during deglaciation events. Also, starting TI at 23 ka and not 24 ka cuts off H2 from Figure 4. Then both deglaciations would have two Heinrich events in the figure. Seems arbitrary to start at 23 and have that white space on the right of the plot. The label on top of LGM in the rectangle shows the state of land ice/sea level from LR04.

Response: Thank you so much for your comments. We modified figures 3 and 4 according to your suggestions and they look better. We start the sequences from glacial maximums and start TI at 24 ka. Attached are the revised figures below.

Figure 3

Figure 4

4) It would be very informative for the reader to label the arrows/lines in figure 5 with some sort of nomenclature to say exactly which of the time series are being used to come up with those bullets. Currently, Figure 5 reads much more like a story rather than a logical argument. By tying these arrow bulleted points to actual points in the time series in Figures 2-4, the reader could follow along the mechanism with quantitative time series. This was partially done in the right panel of figure 6 of the supplementary information by labeling the different events on the second time scale on the right, such that one can partially follow along with the color-coded event and the color-coded time series. However, it is a little difficult to go back and forth between them just based on the color coding, and some sort of other indicator would be more helpful for the reader to follow along through the mechanism that the authors are proposing in Figure 5 of the main manuscript.

Response: Thank you so much for your comments. Considering the readability, we deleted Figure 5.

5) Some of the arrows in the figures are misleading, and I see no reason as to why they're there. For example, the two gray arrows pointing downwards on the right panel of F&G in Figure 4. The Gray arrow in the curve during the LGM at approximately 17.5 ka. This centennial scale oscillation in the record looks much like many other ones and there's no reason to call that particular one out other than the authors subjective selection. I see no quantifiable changes in the record to support it unlike the record from PGM where at 136 ka there a change in the average values during that step down phase showing a clear change in the record. Only use arrows when the math says they should be there - not to try and convince the reader to 'see' something.

Response: Thank you so much for your suggestions. We modified Figure 4 according to your suggestions.

Figure 4

REVIEWERS' COMMENTS

Reviewer #1 (Remarks to the Author):

I generally agree with the authors' additional responses, but I think there are too much speculations based on the subjective comparisons between the data presented by the authors and others. I think the authors' claims through these comparisons are not appropriate to be treated as important content in this paper, and if this part is greatly reduced and the expression is refined, I believe it can be published in Nature Communications.

Response: Thank you so much for your comments and approval. According to your suggestion, we removed sentences including “which could ...” from the section “Different routes to ice age terminations”. Please see in lines 215 to 216:

The H12 event occurred around the boreal summer insolation minimum (Fig. 4a), and created small, partially deglaciated ice sheets prior to TII⁴⁶.

and in lines 217-219:

Although not as intensive as MWP-2B which contributed ~70% of deglacial sea level rise¹¹, MWP-2A was the early phase of ice-sheet retreat before TII and contributed about 30 meters of sea level rise^{49,50}.

We removed this sentence below:

~~Nearly synchronous changes in Antarctic dust flux and atmospheric CO₂ from ~139±1 to 135.7±1 ka BP imply that changes in biological pump and net primary productivity in the Southern Ocean were tightly coupled and already began before the onset of TII.~~

We also modified the last paragraph by removing the discussion over the locations of the early warning signals as below in lines 239-243:

From the viewpoint of physical mechanisms, the abrupt rise of CO₂ since 135.7±1 ka BP was likely due to the increased carbon storage in the stratified Southern Ocean during the preceding PGM⁵⁶. The early warning signal in CO₂ implies that a slow, initial process is important for the shift from glacial to interglacial conditions^{39,57}, which might operate through a way similar to that proposed by Zhang et al.⁵⁸.

Reviewer #3 (Remarks to the Author):

In general, they did a good job at responding to the reviews. Nice update on the figures. I appreciate it that they removed all mention of “the pause”, as the uncertainties in the records did not support that argument. Along the same line of logic, two of the dotted lines in Figure 2 are also not supported by uncertainties in the age models. They are cherry picked for sure. One would be hard pressed to get an objective scientist not associated with the work to look at the time series and draw those connected lines. I am referring to lines 146-148 in the revised manuscript: “Two moderate excursions in the SST record could also correlate to millennial-scale moderate monsoon shifts at approximately 137.5 ± 0.4 and 134 ± 0.6 ka BP.” I recommend that the lines be removed, but the authors are free to say this in text form – although it is not supported by the data.

Response: Thank you so much for your comments. According to your suggestion, we removed this sentence and also removed two dashed lines in Figure 2.

I see where the authors are going with the whole ‘tipping point’ theory, as a gradual increase in insolation and CO₂ cannot explain the abrupt changes during the deglaciation. There are lots of missing dominos in the deglaciation not included in this manuscript, but it is not a comprehensive review. One wording to change would be ‘element’ in line 169. Technically, CO₂ is referred to as a forcing, not an element, with regards to climate. Please update the text accordingly.

Response: Thank you so much for your comments. Updated accordingly. Please see in line 186:

...important tipping forcing for ice age terminations³⁹.

Other than these small changes, the manuscript is ready for print.

Response: Thank you so much for your approval.